# PPAR Gamma Receptor: A Novel Target to Improve Morbidity in Preterm Babies

**DOI:** 10.3390/ph16111530

**Published:** 2023-10-27

**Authors:** Suresh Victor, Ben Forbes, Anne Greenough, A. David Edwards

**Affiliations:** 1Centre for the Developing Brain, Department of Perinatal Imaging and Health, School of Biomedical Engineering and Imaging Sciences, King’s College London, St. Thomas’ Hospital, London SE1 7EH, UK; ad.edwards@kcl.ac.uk; 2Institute of Pharmaceutical Science, King’s College London, Franklin-Wilkins Building, Stamford Street, London SE1 9NH, UK; ben.forbes@kcl.ac.uk; 3Department of Women and Children’s Health, School of Life Course and Population Sciences, King’s College London, Neonatal Intensive Care Centre, Floor 4, Golden Jubilee Wing, King’s College Hospital, Denmark Hill, Brixton, London SE5 9RS, UK; anne.greenough@kcl.ac.uk

**Keywords:** pioglitazone, prematurity, bronchopulmonary dysplasia, lung, brain, adiponectin, PPAR gamma

## Abstract

Worldwide, three-quarters of a million babies are born extremely preterm (<28 weeks gestation) with devastating outcomes: 20% die in the newborn period, a further 35% develop bronchopulmonary dysplasia (BPD), and 10% suffer from cerebral palsy. Pioglitazone, a Peroxisome Proliferator Activated Receptor Gamma (PPARγ) agonist, may reduce the incidence of BPD and improve neurodevelopment in extreme preterm babies. Pioglitazone exerts an anti-inflammatory action mediated through Nuclear Factor-kappa B repression. PPARγ signalling is underactive in preterm babies as adiponectin remains low during the neonatal period. In newborn animal models, pioglitazone has been shown to be protective against BPD, necrotising enterocolitis, and lipopolysaccharide-induced brain injury. Single Nucleotide Polymorphisms of PPARγ are associated with inhibited preterm brain development and impaired neurodevelopment. Pioglitazone was well tolerated by the foetus in reproductive toxicology experiments. Bladder cancer, bone fractures, and macular oedema, seen rarely in adults, may be avoided with a short treatment course. The other effects of pioglitazone, including improved glycaemic control and lipid metabolism, may provide added benefit in the context of prematurity. Currently, there is no formulation of pioglitazone suitable for administration to preterm babies. A liquid formulation of pioglitazone needs to be developed before clinical trials. The potential benefits are likely to outweigh any anticipated safety concerns.

## 1. Introduction

Prematurity accounts for 11% of all births and is the leading global cause of death and disability under five years of age [1]. Bronchopulmonary dysplasia (BPD), also known as chronic lung disease (defined as supplemental oxygen dependency at 36 weeks postmenstrual age), is seen in around 55% of extremely preterm babies. Around 10,000–15,000 new cases are diagnosed annually in the USA and a further 1250 in the UK. Recurrent respiratory symptoms requiring treatment are common, particularly in those who had BPD, even at school age and in adults, as well as lung function abnormalities and exercise intolerance [2]. Recent evidence suggests that affected individuals are at risk of the premature onset of Chronic Obstructive Pulmonary Disease [3]. Nearly 10% of preterm infants suffer from cerebral palsy, with the risk doubling with BPD (OR: 2.10; 95% CI: 1.57–2.82) [4].

Over 30% of survivors following preterm birth experience neurocognitive and socio-emotional problems from early life lasting into adulthood [5]. Psychiatric disorders are present in around 25% of preterm adolescents [6,7]. Prematurity leads to seventy-five million disability-adjusted life years per annum worldwide.

Improvements in mortality and morbidity in preterm babies have so far been achieved through improvements in intensive care management. There is currently no pharmacological treatment available for preterm infants during the newborn period that aims to prevent BPD and improve neurodevelopmental outcomes. Caffeine, when administered to preterm babies, reduced BPD and improved neurodevelopment, demonstrating that pharmacological treatment for neuroprotection is viable postnatally [8]. The CAP trial found a higher patient survival rate without neurodevelopmental disability (cognitive delay, cerebral palsy, severe hearing loss, or bilateral blindness) at a corrected age of 18 to 21 months in infants within the caffeine group compared with those in the placebo group (59.8% vs. 53.8%; adjusted OR: 0.77, 95%CI: 0.64–0.93, *p* = 0.008) [8]. 

There is an urgent need to develop treatment strategies that reduce devastating outcomes in premature babies. We propose that pioglitazone, a Peroxisome Proliferator Activated Receptor Gamma (PPARγ) agonist, may reduce the incidence of death or BPD and may improve neurodevelopment in extreme preterm babies. In this review, we summarise the supporting evidence in non-clinical and clinical studies, the safety considerations, and the pharmaceutical challenges in the translation of this novel treatment strategy.

## 2. Preclinical Studies Using PPARγ Agonists in Preterm Disease Models

PPARγ 1, PPARγ 2, and PPARγ 3 are the three isoforms of PPARγ. Although PPARγ 2 is expressed predominantly in adipose tissue, PPARγ 1 and PPARγ 3 are expressed in adipose tissue, heart, kidney, pancreas, spleen, intestine, colon epithelial cells, and skeletal muscle. In the nervous system, PPARγ is expressed in neurons and glia, including microglia and astrocytes, in multiple brain regions [9,10]. PPARγ is also expressed in lung epithelium, submucosa, and airway smooth muscle [11]. The 15-deoxy prostaglandin J_2_ (15d-PGJ_2_) is a natural ligand for PPARγ, which binds to and activates PPARγ. Thiazolidinediones or glitazones (e.g., pioglitazone and rosiglitazone) are a class of synthetic compounds that function as high-affinity agonists for PPARγ. Studies have been conducted in various animal models of preterm diseases and have shown benefit (Figure 1).

### 2.1. PPARγ Agonists and BPD

PPARγ expression has been reported in several pulmonary cell types, including inflammatory, mesenchymal, and epithelial cells [12]. To define the physiological role of PPARγ within the lung, Simon et al. (2006) generated lung-epithelium-specific PPARγ-deficient mice by breeding the CCtCre transgenic line with mice harbouring loxP sites flanking exon 2 of the PPARγ gene [13]. Airway epithelial cell PPARγ-depleted mice displayed enlarged airspaces, resulting from insufficient postnatal lung maturation. The increase in airspace size was accompanied by alterations in lung physiology, including increased lung volumes and decreased tissue resistance similar to BPD.

Wang et al. (2009) administered either diluent or rosiglitazone (0.1 to 8 mg/kg intraperitoneal once daily for either one day or seven days) to newborn Sprague Dawley rat pups [14]. The pups were then sacrificed, and the lungs were examined for specific markers of alveolar epithelial, mesenchymal, and vascular maturation and lung morphometry. Systemically administered rosiglitazone significantly enhanced lung maturation without affecting serum electrolytes, blood glucose, blood gases, plasma cholesterol, triglycerides, and serum cardiac troponin levels. There was no significant effect on blood glucose at either day one or day seven for any of the doses of rosiglitazone examined, except for the highest dose used, i.e., 8 mg/kg for seven days, which resulted in statistically significantly lower glucose values compared with controls (84 ± 15 mg/dL vs. 127 ± 3 mg/dL, mean ± SD, *p* < 0.05, rosiglitazone versus control). Similar effects were noted when 15d-PGJ_2_ was administered instead of rosiglitazone.

Rehan et al. (2006) exposed newborn Sprague Dawley rat pups to three conditions: control (21% O_2_), hyperoxia alone (95% O_2_ for 24 h), or hyperoxia with rosiglitazone (95% O_2_ for 24 h + rosiglitazone, 3 mg/kg, administered intraperitoneally) [15]. In the lungs of hyperoxic animals, septal thickness was reduced significantly (40% vs. 15%) compared with control lungs (*p* < 0.01, 95% vs. 21% O_2_). Quantitative analysis of the alveolar number demonstrated a significant reduction (50 vs. 15) in the average radial alveolar counts in hyperoxic animals compared with controls. A significant decrease in the expression of lipogenic markers, and a significant increase in the expression of myogenic markers in the hyperoxia-alone group were observed. Exposure to hyperoxia alone markedly decreased PPARγ and markedly increased alpha smooth muscle actin expression. These hyperoxia-induced morphologic, molecular, and immunohistochemical changes were completely prevented by rosiglitazone.

Lee et al. (2020) administered dexamethasone or pioglitazone (two doses via 0.3–3 mg/kg bolus intraperitoneal injection) to pregnant Sprague Dawley rat dams on embryonic day 18 and 19. Their pups were delivered by caesarean section on day 20 [16]. Antenatal pioglitazone increased alveolar epithelial and mesenchymal maturation markers equally in males and females. In contrast, antenatal dexamethasone had sex-specific effects. Dexamethasone (0.25 mg/kg via intraperitoneal injection on embryonic days 18 and 19) significantly increased surfactant phospholipid synthesis and alveolar epithelial and mesenchymal maturation markers in both males and females. However, this increase was more prominent in females. Additionally, unlike dexamethasone, antenatal pioglitazone did not increase hippocampal apoptosis.

Richter et al. (2016) injected pregnant rabbits with saline or rosiglitazone (3 mg/kg) 48 h and 24 h prior to preterm delivery at 28 days of gestation (term = 31 days) [17]. The pups were held in normoxia (21% O_2_) or hyperoxia (>95% O_2_) conditions, and assessment was conducted at three different time points (1 h, 24 h, and 7 days). Rosiglitazone significantly increased the expression of Vascular Endothelial Growth Factor (VEGF) and surfactant protein B immediately after delivery. On day seven, the lung parenchymal architecture was more mature in rosiglitazone-exposed pups.

Morales et al. (2014) administered one-day-old Sprague Dawley rat pups PPARγ agonists rosiglitazone (3 mg/kg), pioglitazone (3 mg/kg), or the diluent via nebulization every 24 h. The animals were exposed to 21% or 95% O_2_ for up to 72 h [18]. Rosiglitazone and pioglitazone enhanced lung maturation in both males and females, as evidenced by the increased expression of markers of alveolar epithelial and mesenchymal maturation at 72 h after initial nebulisation. This approach also protected against hyperoxia-induced lung injury.

In one-day-old rats exposed to 95% O_2_, aerosolized/intraperitoneal pioglitazone (1–3 mg/kg; plasma concentration: 0.52–0.99 ± 0.15 µg/mL) once daily for three days alone or in combination with surfactant showed significantly increased markers of lung maturation, significantly decreased markers of lung inflammation, and prevention of lung injury [18,19].

One-day-old rats induced through the intra-amniotic delivery of lipopolysaccharide (LPS) and postnatal hyperoxia (80% for 7 days) and treated with intraperitoneal rosiglitazone (3 mg/kg for 14 days) had significantly smaller and more complex airspaces, larger alveolar surface area, greater pulmonary vascular density, restored levels of VEGF and its receptor, and decreased right ventricular hypertrophy [20].

### 2.2. PPARγ Agonists and Preterm Brain Injury

Pioglitazone represses two signalling pathways associated with preterm brain injury, (a) the Nuclear Factor-kappa B (NF-κB) signalling pathway, which promotes inflammation mediated injury [21], and (b) the WnT canonical signalling pathway, which inhibits developmental myelination [22]. The NF-κB activation in microglia plays a significant role in the pathogenesis of hypoxic ischemic injury of the immature brain, and its prophylactic inhibition offers significant neuroprotection [23]. PPARγ agonists also protect oligodendrocyte progenitors against maturational arrest induced by the inflammatory cytokine Tumor Necrosis Factor–alpha (TNF-α) by affecting mitochondrial function [24].

Pioglitazone (20 mg/kg via a single intraperitoneal dose) significantly improved LPS-induced neuro-behavioural and physiological disturbances, including the loss of weight, hypothermia, the righting reflex, the wire hanging manoeuvre, negative geotaxis, and hind limb suspension, in postnatal day five rats [25]. Pioglitazone attenuated the LPS-induced loss of oligodendrocytes, the reduction of mitochondrial complex one activity, increases in thiobarbituric-acid-reactive substance content, and increases in microglial activation and inflammatory responses. Pioglitazone attenuated the LPS-induced increase of IL (Interleukin)-1β in both serum and the brain (*p* < 0.05) and increases in IL-6 and TNF-α levels in the serum of rats (*p* < 0.05) [25].

Krishna et al. (2021) studied the immunoreactivity of PPARγ in microglia, oligodendrocyte progenitor cells, and astrocytes in postmortem samples from the frontoparietal cortex of preterm infants (23–28 weeks of gestational age) with and without intraventricular haemorrhage (IVH) [26]. PPAR-γ was abundantly expressed by ionized calcium binding adaptor molecule 1 (Iba1) positive microglia/macrophages in the periventricular white matter and ganglionic eminences of infants with IVH, but scarcely in infants without IVH. Accordingly, quantification showed that both total Iba1- and PPARγ-Iba1-positive cells were significantly greater in density in infants with IVH compared with controls without IVH (*p* < 0.001 for both).

Krishna et al. (2021) also used intraperitoneal glycerol to induce IVH in E29 rabbit kits (term E32) at 3 to 4 h of age. The kits with IVH were treated with either intramuscular rosiglitazone (0.1 mg/kg twice a day for seven days, starting on day 1) or a vehicle [26]. Treatment with rosiglitazone further elevated PPARγ levels in microglia, reduced proinflammatory cytokines, increased microglial phagocytosis, and improved oligodendrocyte progenitor cell maturation. PPARγ activation enhanced myelination and neurological function in kits with IVH. Kits treated with rosiglitazone were significantly more active for a longer time compared with vehicle controls (*p* = 0.048). The distance travelled and speed within both the entire and central arena showed a trend toward an increase in rosiglitazone-treated kits compared with vehicle controls; however, this comparison was not statistically significant.

### 2.3. PPAR Agonists and Necrotising Enterocolitis

Swiss Webster mice were randomized to receive sham (control) or ischemia/reperfusion injury to the gut, which was induced by the transient occlusion of the superior mesenteric artery for 45 min with variable periods of reperfusion. Ischemia/reperfusion injury resulted in the early induction of PPARγ expression and the activation of NF-κB in the small intestine. Pre-treatment with a PPARγ agonist, 15d-PGJ_2_ (2 mg/kg), 45 min prior to injury attenuated the intestinal NF-κB response and ischemia/reperfusion induced gut injury. The activation of PPARγ demonstrated a protective effect on the small bowel during ischemia/reperfusion-induced gut injury [27].

Corsini et al. (2016) studied preterm rats in which necrotising enterocolitis was induced using the hypoxia–hypothermia model [28]. The treatment group (*n* = 30) received enteral pioglitazone (10 mg/kg/day) for 72 h and the control group (*n* = 30) did not. The animals were sacrificed 96 h after birth. Necrotising enterocolitis was diagnosed by evaluating histological ileum changes, and messenger RNA (mRNA) levels of IL-4, IL-12, IL-6, IL-10, Interferon-γ, and TNF-α cytokines were measured. Necrotising enterocolitis occurrence was higher in the control group (18/30; 60%) than in the treatment group (5/30; 16.7%) and was more severe. Proinflammatory IL-12 and Interferon-γ mRNA levels were significantly lower in the treatment group than in the control group. Conversely, anti-inflammatory IL-4 mRNA levels were significantly higher in the treatment group than in the control group.

## 3. Clinical Studies on PPARγ Activity in Premature Babies

### 3.1. Adiponectin Concentrations in Premature Babies

Adiponectin is a robust biomarker of PPARγ activity [29]. The existing literature on plasma adiponectin concentrations in preterm newborn babies suggests that PPARγ signalling may be underactive in preterm newborn babies. Kajantie et al. (2004) measured plasma adiponectin concentrations using ELISA in the cord vein of 197 infants [30]. Of them, 122 were born preterm (22 to 32 weeks gestation) and 75 at term (49 from a healthy pregnancy and 26 from a diabetic pregnancy with similar findings, and thus all data from term infants were pooled). At birth, preterm babies have low plasma adiponectin concentrations (*n* = 122; 3.7 ± 10.6 μg/mL) when compared to term babies (*n* = 75; 33.7 ± 13.6 μg/mL) [30]. Mean adiponectin concentrations increased from less than 1 μg/mL at 24 weeks of gestation to approximately 20 μg/mL at term. Preterm females had 57% higher adiponectin concentrations (0 to 146%; *p* = 0.05) than preterm males. Adiponectin levels were lower in preterm infants with recent (<12 h) exposure to maternal betamethasone, but were unrelated to the mode of delivery, preeclampsia, or impaired umbilical artery flow [30].

Hansen-Pupp et al. (2015) analysed adiponectin concentrations in cord blood at birth and peripheral blood at 72 h, on day 7, and then weekly until the postmenstrual age of 40 weeks in 52 preterm babies born at 26 ± 1.9 weeks gestational age [31]. The mean adiponectin concentration increased from 6.8 ± 4.4 μg/mL at 72 h to 37.4 ± 22.2 μg/mL at three weeks. The mean adiponectin concentration during days 3 to 21 (21.4 ± 12 μg/mL) correlated with gestational age at birth (r = 0.46, *p* = 0.001), birth weight (r = 0.71, *p* = 0.001), and birth weight Standard Deviation Score (SDS) (r = 0.42, *p* = 0.003). Furthermore, the mean adiponectin concentration during days 3 to 21 correlated with weight SDS, length SDS, and head circumference SDS at 35 weeks corrected gestational age (r = 0.62, 0.65, and 0.62, respectively; all *p* = 0.001). Peak concentrations at 3 weeks of age (*n* = 52; 37.4 ± 22.2 μg/mL) did not correlate with gestational age, but positively correlated with catch-up growth (β = 0.021, CI:0.001–0.041, *p* = 0.04) [31].

### 3.2. PPARγ Signalling in Brain Imaging Genetics Studies

In a candidate gene study of thirteen genes from seventy-two preterm infants, Single Nucleotide Polymorphisms (SNPs) in the genes FADS2 and ARVCF were significantly associated with fractional anisotropy (FA) in white matter extracted using Tract Based Spatial Statistics (TBSS). FADS2 is involved in lipid metabolism, including PPARγ signalling [32]. A pathway-based genome-wide imaging genomics analysis was carried out on the same cohort, using the Pathways sparse Reduced Rank Regression (sRRR) machine learning approach with genome-wide SNP (Single Nucleotide Polymorphism) genotyping and a reduced version of the same TBSS phenotype. This showed that the PPARγ signalling pathway was the top ranked pathway in the model, which included adjustment for both gestational age and post menstrual age [33].

A SNP-based genome-wide imaging genomics analysis was carried on a large independent cohort of 271 preterm infants, using the sRRR method with genome-wide SNP genotyping and a probabilistic tractography phenotype incorporating FA. This detected an association between SNPs in the PPARγ gene, and the imaging phenotype was fully adjusted for gestational age, post menstrual age, and ancestry [34].

Meirhaeghe et al. (2007) genotyped two independent cross-sectional studies from Northern Ireland (*n* = 382 and 620) for the Pro12Ala polymorphism of PPARγ2 [35]. In combined populations, the PPARγ2 Ala12 allele was associated (*p* = 0.03) with lower birth weight, primarily caused by shorter gestational duration (*p* = 0.04). The frequency of Ala12 allele carriers was higher (*p* = 0.027) in the group of individuals born before term (35%, *n* = 60) than in the group of individuals born at term (22%, *n* = 942). The odds ratios (95% CI) of preterm birth for Ala12 allele carriers were 1.9 (1.1–3.4), *p* = 0.022, and 4.2 (1.9–9.7), *p* = 0.0006 (adjusted for sex, maternal age, and study), when considering 37 or 35 weeks of pregnancy as a threshold for preterm birth, respectively.

The association between PPARγ2 Pro12Ala polymorphisms and neurodevelopment at 18–24 months of age was assessed in two groups of European infants (155 born before 33 weeks of gestation and 180 born later than 36 weeks of gestation) [36]. The Ala allele of the Pro12Ala polymorphism was noted in 25% of the preterm infants and 20% of the term infants. The Ala allele of PPARγ2 was significantly associated with adverse cognitive (*p* = 0.019), language (*p* = 0.03), and motor development (*p* = 0.036) at 18 to 24 months of age after taking into consideration the duration of ventilation, gender, and index of multiple deprivation scores, but without correction for potential shared ancestry. There was no association between the PPARγ2 Pro12Ala polymorphism and neurodevelopment in term infants.

### 3.3. Clinical Trials of PPARγ Agonists in Preterm Babies

No clinical trials have been conducted using PPARγ agonists in preterm babies. Pioglitazone has been trialled in children and adults for its neuroprotective action and has been found to be safe. Improvements in behaviour were demonstrated in 4 to 12-year-old children (*n* = 44; 30 mg once daily) with Autistic Spectral Disorders after 10 weeks of treatment with pioglitazone [37]. The occurrence of adverse events was mild and transient, and none warranted medical intervention or alteration of the treatment regimen. Vomiting (4% vs. 3% in controls) and headache (3% vs. 3% in controls) were the most frequent side-effects reported in the pioglitazone group (*n* = 22).

## 4. Safety Considerations for Using Pioglitazone in Premature Babies 

The effects of pioglitazone on preterm lung disease have been demonstrated in animal models at plasma concentrations of 0.52–0.99 ± 0.15 µg/mL [18]. This target plasma concentration can be achieved at dose ranges currently licensed in adults for diabetic treatment. Efficacy in animal experiments has also been demonstrated with a short course of treatment lasting up to 14 days [20]. Studies using samples of airway secretions have demonstrated that lung inflammation peaks between 7 and 10 days after birth in premature babies and is exacerbated by both antenatal and nosocomial infections [38]. Hence, the duration of treatment for two to three weeks starting from soon after birth may be sufficient for the prevention of BPD in preterm babies. A much shorter duration of treatment in comparison to the current licensed treatment for adult diabetes needs to be taken into consideration when evaluating the safety of pioglitazone in premature babies.

### 4.1. Increased Insulin Sensitivity

Pioglitazone hydrochloride is an antidiabetic agent that depends on the presence of insulin for its mechanism of action. Pioglitazone decreases insulin resistance in the peripheral muscle and liver, resulting in increased insulin-dependent glucose disposal and decreased hepatic glucose output, respectively. It is not an insulin secretagogue [39].

Hyperglycaemia is often seen in premature newborn babies and is likely to be due to decreased insulin production. Since pioglitazone enhances the effects of circulating insulin (by decreasing insulin resistance), it does not lower blood glucose in animal models that lack endogenous insulin. Therefore, hypoglycaemia is not expected in preterm babies when treated with pioglitazone alone. However, there is a risk of hypoglycaemia in premature babies when combined with insulin treatment. Since the threshold for starting insulin treatment in premature babies is set around 12 mmol/L and hypoglycaemia is defined as less than 2.5 mmol/L, there is a sufficient safety margin to titrate the insulin treatment to maintain normoglycemia.

### 4.2. Effect on Lipid Metabolism

In pharmacodynamic studies of both monotherapy and combination therapy, treatment with pioglitazone was associated with decreased blood free fatty acids and triglyceride concentrations and increased blood High-Density Lipoprotein Cholesterol (HDL-C) concentrations [39]. Preterm infants on total parenteral nutrition often have increased level of blood triglyceride concentrations resulting in the temporary cessation of lipid supplementation until triglyceride levels decrease [40]. A reduction in blood triglyceride concentrations with pioglitazone treatment may be beneficial for preterm babies, with improved lipid metabolism resulting in improved weight gain.

### 4.3. Weight Gain

Pioglitazone is associated with weight gain after 24 weeks of treatment, possibly due to oedema [39]. In addition, pioglitazone significantly decreases visceral (abdominal) fat stores while increasing extra-abdominal fat. The reduction in visceral fat correlates with improved hepatic and peripheral tissue insulin sensitivity.

In preterm babies, peak adiponectin concentrations on day 21 after birth were associated with increased discharge weight, suggesting that weight gain may be due to reasons other than oedema in preterm babies [31]. Concerns related to excess weight gain or oedema may not be an issue with a short duration of treatment.

### 4.4. Rare Side Effects

Bladder cancer, fractures in women, and macular oedema have rarely been reported in adult diabetic patients [39]. These have been associated with prolonged treatment (>1 year). Rare transient elevations in liver enzymes and very rare clinically apparent liver injury have been noted.

### 4.5. Reproduction Toxicity in Animals

Several animal studies have been conducted by Takeda Pharmaceuticals for the licensing of pioglitazone [39]. Pioglitazone is non-teratogenic in both rats and rabbits. High doses were associated with maternal toxicity (e.g., placental pathology and weight loss), implantation losses, reduced foetal size, live birth weight and survival ratios, and delayed pup development. Foetal growth restriction was attributable to the action of pioglitazone in diminishing the maternal hyperinsulinemia and increased insulin resistance that occurs during pregnancy, thereby reducing the availability of metabolic substrates for foetal growth rather than a direct toxic effect.

Six pregnant rats per group received twice daily oral doses of 0, 50, 100, 200, or 400 mg/kg of pioglitazone on gestational days six through to twenty. Based on the results of this study, the no observable adverse effect level (NOAEL) was determined to be less than 50 mg/kg of pioglitazone.

Thirty-three to thirty-five rat pups received enteral doses of 0, 20, 40, or 80 mg/kg/day of pioglitazone on gestational days six through to seventeen. The NOAEL for developmental toxicity was determined to be 20 mg/kg based on the reductions in embryo viability, litter size, and pup survival in the 40 mg/kg group. The NOAEL for F_1_ generation rats was 40 mg/kg based on reduced body weights for males and reduced body weight gains (gestational period) for females in the 80 mg/kg group.

Five pregnant rabbits received oral doses of 0, 5, 40, 100, or 400 mg/kg/day of pioglitazone on gestational days six through to eighteen. The top dose was reduced to 160 mg/kg due to high mortality. Based on the results of this study, the NOAEL for maternal and developmental toxicity was determined to be closer to 40 mg/kg.

Fifteen to eighteen rabbits were given enteral doses of 0, 40, 80, or 160 mg/kg/day of pioglitazone on gestational days six through to eighteen. Mild signs of maternal toxicity were noted at 80 and 160 mg/kg/day. The NOAEL for maternal toxicity was 40 mg/kg/day. Slight embryotoxicity was noted at 160 mg/kg/day.

## 5. Pharmaceutical Challenges in Creating a Formulation Suited for Premature Babies

Pioglitazone is currently available as a licensed medicine in tablet form. A formulation suitable for administration to premature babies is not commercially available. The major challenge for developing a liquid formulation is that pioglitazone freebase and its hydrochloride salt, which is used in pharmaceutical products, have extremely low aqueous solubilities (<1 mg/mL). In animal experiments, pioglitazone has been dissolved in surfactant. This limits the duration of treatment as usually not more than two doses of surfactant are administered in preterm babies. Several solubility studies on pioglitazone have been reported. Most of the excipients used in these studies are not suitable for use in preterm babies [41,42,43,44]. The choice of excipients is restricted by the limited experience in the safety profiles of excipients in newborn babies and children. Furthermore, the concentration of these excipients cannot exceed the daily consumption limits set by regulatory bodies.

Creating a solution formulation of pioglitazone at a concentration that enables delivery of the dose within the fluid restrictions for preterm babies is a pharmaceutical challenge, especially with the limited portfolio of co-solvents and solublisers that have regulatory acceptance for use in children. The pioglitazone concentration must be such that dosing per kilogram body weight in extremely low birth weight babies (<1000 g birth weight) is not impractical due to too small measurable volumes and not too large a volume that bigger babies (>1500 g birth weight) cannot tolerate. This applies to both intravenous and oral preparations.

Different routes of administration, such as intravenous (injection), endotracheal (tube), or inhalation (nebuliser), can be considered in the newborn period when infants are not tolerating enteral feeds. Oral formulations, e.g., solutions or suspensions, can be used for subsequent maintenance treatment when enteral feeds are tolerated. When considering oral formulations, such as suspensions, the viscosity of the formulation and particle size needs to be sufficiently low to allow for administration through size five French gastric tubes.

Considerations for liquid pharmaceutical formulations include chemical and microbiological stability. The chemical stability of pioglitazone needs to be verified to remain within pharmacopeial limits for the developed formulation when stored in its final container under specified conditions, e.g., room or fridge temperature. Preservatives are normally required to assure the microbiological quality of multiple-dose liquid preparations. However, for many preservatives there is still limited data regarding the levels of safe exposure in children of different ages. The need to preserve a paediatric preparation and the choice of the preservative system at the lowest concentration feasible should be justified in terms of benefit–risk balance.

## 6. Future Perspectives and Scope

There is an urgent need to develop treatment strategies that reduce devastating outcomes in premature babies. We propose that a PPARγ agonist may reduce the incidence of death or BPD and improve neurodevelopment in extreme preterm babies. The availability of a safe, well-tolerated agonist for PPARγ signalling (pioglitazone) allows the exploration of a repurposing strategy to test this approach in the clinic. We are currently developing formulation options for pioglitazone that will be suitable for administration to preterm babies. This will allow us to conduct early-phase clinical trials to determine the safety, tolerance, and pharmacokinetics of pioglitazone in preterm newborn babies. In an online UK survey conducted by us in January 2023 of 207 parents who had a premature baby born before 28 weeks of gestational age, 94% agreed that preventing lung and brain disease is an important treatment outcome for preterm babies. Ninety-six percent stated they would have agreed to participate in a randomised, placebo controlled, early-phase clinical trial of a prophylactic treatment even if their child may not have developed the disease. Furthermore, 74% said that the potential side effects of the treatment (e.g., oedema, abnormal liver function, and an increased risk of hypoglycaemia) would not have worried them.

## 7. Conclusions

Prematurity has a devastating impact on individuals, families, and society. There is an urgent need to develop treatment strategies that improve lung and brain outcomes in premature newborn babies. Pioglitazone, a PPARγ agonist, may reduce the incidence of death or BPD and improve neurodevelopment in extreme preterm babies. A stable liquid formulation of pioglitazone suitable for preterm babies needs to be developed before this treatment can progress to the clinic. The potential benefits are likely to outweigh any anticipated safety concerns.

## Figures and Tables

**Figure 1 pharmaceuticals-16-01530-f001:**
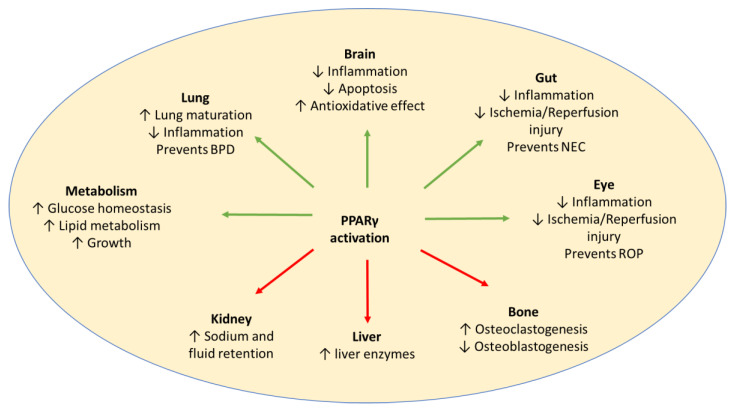
Activation of PPARγ may have potential beneficial effects (green arrows), as well as adverse effects (red arrows), in preterm babies. Adverse effects may be avoided by a short duration of treatment. BPD: Bronchopulmonary Dysplasia; NEC: Necrotising Enterocolitis; ROP: Retinopathy of Prematurity.

## Data Availability

Data sharing is not applicable.

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
