# Peer review of "PPAR Gamma Receptor: A Novel Target to Improve Morbidity in Preterm Babies"

_pharmaceuticals, 2023, doi:10.3390/ph16111530_

Round 1
Author Response
In the article … Victor et al make the case for further research into the potential of PPAR-γ receptor agonists in the prevention of morbidity in pre-term born neonates caused by chronic inflammation. The article is well written and gives a good insight into what is currently known about these compounds from animal studies. There is also a prominent place for potential toxic effects of the compounds, although most are considered not relevant by the authors.
We thank the reviewer for their comments and have responded to them as follows:
Starting at the safety considerations I think that the statement of the others that hypoglycemia is not often seen in premature infants is not true. Premature infants very frequently show hypoglycemia, one of the reasons they are started in iv fluids directly after birth in some centers preferably from stabilization directly after birth. So this is a very important aspect to monitor. Especially since the long-term effects of these hypoglycemic events are unknown.
Response:
We have stated the following: Hyperglycaemia is often seen in premature newborn babies and is likely to be due to decreased insulin production.
We have already stated that there is a risk of hypoglycaemia when pioglitazone treatment is combined with insulin. Monitoring blood glucose concentrations post administration of pioglitazone is important particularly when insulin is being used.
We agree that both hyperglycaemia and hypoglycaemia can be seen in preterm babies. Our intention is to say that pioglitazone is unlikely to make either worse or more frequent.
These long-term effects in general should be mentioned as well. The current safety overview only states short-term risks. However, introducing a preventive compound in neonates askes for a very thorough balance of potential benefits and side-effects, including long-term effects. Therefor performance during follow-up years after birth should be a primary outcome, rather than just the short-term of NICU admission. It takes time for neurodevelopmental delays to become apparent.
Response:
We have already stated: We propose that a PPARγ agonist may reduce the incidence of death or BPD and improve neurodevelopment in extreme preterm babies.
We have already considered long term neurodevelopment as an important outcome measure.
In the article mention the trial of pioglitazone in children with ASD. Looking at that trial is becomes clear that a substantial part of the participants suffered from headache and vomiting. This is an important factor that is now underestimated. Feeding is one of the most important part of care for premature infants. Feeding problems may lead to longer parenteral feeding with increased risk of invasive infections that are correlated with increased risk for the morbidities that are to be prevented by the PPAR γ receptor agonists. This potential concern should be part of the overview that is given by the authors.
Response:
Although the trial participants most commonly experienced vomiting and headache, this was still an uncommon side effect and not significantly more than the control group.
The following statement has been added:
Occurrence of adverse events was mild and transient, and none warranted medical intervention or alteration of the treatment regimen. Vomiting (4% vs. 3% in controls) and headache (3% vs. 3% in controls) were the most frequent side-effects reported in the pioglitazone group (n = 22).”
In addition to the studies mentioned in the current version of the manuscript another paper discusses the effect of pioglitazone in reducing the effects of IVH in rabbits. (PMID: 34462350). This is worth mentioning.
Response: The following additional paragraphs and references have been included:
Line 153-155:
PPAR-γ agonists also protect oligodendrocyte progenitors against the maturational arrest induced by the inflammatory cytokine Tumor Necrosis Factor - alpha (TNF-α) by affecting mitochondrial functions .
De Nuccio, C.; Bernardo, A.; Cruciani, C.; De Simone, R.; Visentin, S.; & Minghetti, L. Peroxisome proliferator activated receptor-γ agonists protect oligodendrocyte progenitors against tumor necrosis factor-alpha-induced damage: Effects on mitochondrial functions and differentiation. Experimental Neurology 2015, 271, 506–514. https://doi.org/10.1016/j.expneurol.2015.07.014
Lines 165- 184:
Krishna et al. (2021) studied the immunoreactivity of PPAR-γ in microglia, oligodendrocyte progenitor cells, and astrocytes in postmortem samples from the frontoparietal cortex of preterm infants (23 - 28 weeks of gestational age) with and without intraventricular haemorrhage (IVH) [25]. PPAR-γ was abundantly expressed by Ionized calcium binding adaptor molecule 1 (Iba1) microglia or macrophages in the periventricular white matter and ganglionic eminences of infants with IVH but scarcely in infants without IVH. Accordingly, quantification showed that both total Iba1 positive and Iba1 and PPARγ positive cells were significantly greater in density in infants with IVH compared with controls without IVH (P < 0.001 for both).
Krishna et al. (2021) also used intraperitoneal glycerol to induce IVH in E29 rabbit kits (term E32) at 3 to 4 h of age, and then the kits with IVH were treated with either intramuscular rosiglitazone (0.1 mg/kg twice a day for seven days, starting at day 1) or vehicle [25]. Treatment with rosiglitazone further elevated PPARγ levels in microglia, reduced proinflammatory cytokines, increased microglial phagocytosis, and improved oligodendrocyte progenitor cell maturation. PPARγ activation enhanced myelination and neurological function in kits with IVH. Kits treated with rosiglitazone were significantly more active for a longer time compared with vehicle controls (P = 0.048). The distance travelled and speed within both the entire and central arena showed a trend toward increase in rosiglitazone-treated kits compared with vehicle controls; however, the comparison was not statistically significant.
[25] Krishna, S.; Cheng, B.; Sharma, D. R.; Yadav, S.; Stempinski, E. S.; Mamtani, S.; Shah, E.; Deo, A.; Acherjee, T.; Thomas, T.; et al. PPAR-γ activation enhances myelination and neurological recovery in premature rabbits with intraventricular hemorrhage. Proceedings of the National Academy of Sciences of the United States of America 2021, 118, e2103084118. https://doi.org/10.1073/pnas.2103084118
Reviewer 2 Report
The complications associated with preterm birth remain problematic, and even more so as a higher percentage of babies survive birth at less than 28 weeks of gestation. These extremely low gestational age neonates (ELGAN) are at higher risk of BPD, NEC, compromised neurodevelopment, and other morbidities. Only time will tell what other organs are compromised with later morbidities. The possible benefits of PPARγ therapy for preterm infants address a critical need for interventions. The following comments are provided as suggestions the authors may address to strengthen their contribution.
1. The use of caffeine as an example of an intervention for reducing BPD is misleading. The benefit of caffeine is the reduction of apnea, which would lead to the use of conventional mechanical ventilation known to cause lung injury that precipitates BPD. It is well established that BPD is associated with compromised neurodevelopment and development of other organs. As of now, there are no drugs that are recognized to address BPD pathogenesis. PPARγ might be a candidate to reduce the adverse impacts, which is the focus of this review. If successful, this would be valuable for reducing BPD comorbidities.
2. Reference 18 used appropriate controls and provided some insights into the responses of term rat pups to PPARγ agonists not exposed to hyperoxia known to induce BPD pathogenesis. Unfortunately, assessing the responses after preterm delivery is limited with rats. It remains uncertain how PPARγ agonists influence development after uncomplicated preterm birth.
3. Linea 163-170. This section, though interesting, does not seem relevant to prematurity. Ref 25 used 6-week old mice and Ref 26 queried candidate genes. Consider deleting or justifying the relevance to avoid criticism.
4. Line 193. Provide a reference for adiponectin as marker for PPARγ.
5. Lines 199-200. Provide a p-value.
6. Lines 201,202,209,210. g/ml should be ug/ml. Please make sure all such values are correct throughout.
7. Lines 206-217. Did adiponectin increase between what was measured in cord blood and at 72 h? Are there any data for postnatal changes in PPARγ among term and preterm infants?
8. Sections 4.2 and 4.3. Preterm infants are at risk of increased adiposity that is associated with increased risk of later obesity. Weight gain due to increased fat deposition will not be considered by many to be beneficial. Increasing fat-free mass would be better.
9. Section 4.5. Are there any data for the maternal-to-fetal transfer of pioglitazone? Such studies would need to use animal models with placental structures similar to those of humans. If PPARγ agonists are not transferred across the placenta to the fetus, the adverse response would be of maternal origin.
10. Lines 339-340. This repeats lines 314-315.
11. Section 5. A key consideration for the preterm infant is organ immaturity that will influence metabolism, distribution, responses, and elimination of pioglitazone and other PPARγ agonists. Assessing those will require animal models that are more relevant to ELGAN that rodents.
12. Section 6. The authors should consider mentioning the limitations of previous studies, such as the animal models, and suggest recommendations for what is needed to translate findings into clinical trials. This might also help to position them for such studies.
Author Response
Reviewer 2:
We thank the reviewer for their comments and have responded to them as follows:
The complications associated with preterm birth remain problematic, and even more so as a higher percentage of babies survive birth at less than 28 weeks of gestation. These extremely low gestational age neonates (ELGAN) are at higher risk of BPD, NEC, compromised neurodevelopment, and other morbidities. Only time will tell what other organs are compromised with later morbidities. The possible benefits of PPARγ therapy for preterm infants address a critical need for interventions. The following comments are provided as suggestions the authors may address to strengthen their contribution.
- The use of caffeine as an example of an intervention for reducing BPD is misleading. The benefit of caffeine is the reduction of apnea, which would lead to the use of conventional mechanical ventilation known to cause lung injury that precipitates BPD. It is well established that BPD is associated with compromised neurodevelopment and development of other organs. As of now, there are no drugs that are recognized to address BPD pathogenesis. PPARγ might be a candidate to reduce the adverse impacts, which is the focus of this review. If successful, this would be valuable for reducing BPD comorbidities.
Response: The following change has been made:
There is currently no pharmacological treatment available for preterm infants during the newborn period that aims to prevent BPD and improve neurodevelopmental outcomes. Caffeine, when administered to preterm babies reduced BPD and improved neurodevelopment demonstrating that pharmacological treatment for BPD and neuroprotection is viable postnatally [8]. The CAP trial found a higher patient survival rate without neurodevelopmental disability (cognitive delay, cerebral palsy, severe hearing loss or bilateral blindness) at a corrected age of 18 to 21 months in infants within the caffeine group compared with those in the placebo group (59.8% versus 53.8%; adjusted OR: 0.77, 95%CI: 0.64-0.93, P = 0.008) [8].
- Reference 18 used appropriate controls and provided some insights into the responses of term rat pups to PPARγ agonists not exposed to hyperoxia known to induce BPD pathogenesis. Unfortunately, assessing the responses after preterm delivery is limited with rats. It remains uncertain how PPARγ agonists influence development after uncomplicated preterm birth.
Response:
The data on lung injury is not exclusive to rat models. Richter et al, used a preterm rabbit model, and demonstrated that prenatal maternal administration of rosiglitazone attenuates neonatal hyperoxic lung injury and results in a more mature pulmonary parenchyma.
As for brain injury, please see new reference added in response to comments from the other reviewer. The following has been added.
Krishna et al. (2021) studied the immunoreactivity of PPAR-γ in microglia, OPCs, and astrocytes in postmortem samples from the frontoparietal cortex of preterm infants (23 - 28 weeks of gestational age) with and without intraventricular haemorrhage (IVH). PPAR-γ was abundantly expressed by Iba1+ microglia/macrophages in the periventricular white matter and ganglionic eminences of infants with IVH but scarcely in infants without IVH. Accordingly, quantification showed that both total Iba1+ and Iba1+PPAR-γ+ cells were significantly greater in density in infants with IVH compared with controls without IVH (P < 0.001 for both).
Krishna et al. (2021) also used intraperitoneal glycerol to induce IVH in E29 rabbit kits (term E32) at 3 to 4 h of age, and then the kits with IVH were treated with either intramuscular rosiglitazone (0.1 mg/kg twice a day for seven days, starting at day 1) or vehicle. Treatment with rosiglitazone further elevated PPAR-γ levels in microglia, reduced proinflammatory cytokines, increased microglial phagocytosis, and improved oligodendrocyte progenitor cell maturation. PPAR-γ activation enhanced myelination and neurological function in kits with IVH. Kits treated with rosiglitazone were significantly more active for a longer time compared with vehicle controls (P = 0.048). The distance travelled and speed within both the entire and central arena showed a trend toward increase in rosiglitazone-treated kits compared with vehicle controls; however, the comparison was not statistically significant.
[25] Krishna, S.; Cheng, B.; Sharma, D. R.; Yadav, S.; Stempinski, E. S.; Mamtani, S.; Shah, E.; Deo, A.; Acherjee, T.; Thomas, T.; et al. PPAR-γ activation enhances myelination and neurological recovery in premature rabbits with intraventricular hemorrhage. Proceedings of the National Academy of Sciences of the United States of America 2021, 118, e2103084118. https://doi.org/10.1073/pnas.2103084118
- Linea 163-170. This section, though interesting, does not seem relevant to prematurity. Ref 25 used 6-week-old mice and Ref 26 queried candidate genes. Consider deleting or justifying the relevance to avoid criticism.
Response:
Section deleted
Retinal ischemia/reperfusion injury is considered to be associated with retinopathy of prematurity (ROP). In a mouse model of retinal ischemia/reperfusion, pioglitazone promoted the survival of retinal cells in ganglion cell layer following retinal ischemia/reperfusion injury (P< 0.05) by suppressing ischemia/reperfusion-induced activation of Toll-like receptor 4 and NLR family pyrin domain containing 3 (NLRP3) inflammasomes via inhibiting NF-κB and p38 phosphorylation [25]. Pioglitazone was one of the novel candidate compounds identified for the treatment of ROP using computational drug-gene association analysis [26].
- Line 193. Provide a reference for adiponectin as marker for PPARγ.
Response: Reference added
Adiponectin is a robust biomarker of PPARγ activity [28].
[28] Wagner, J. A.; Wright, E. C.; Ennis, M. M.; Prince, M.; Kochan, J.; Nunez, D. J. R.; Schneider, B.; Wang, M. D.; Chen, Y.; Ghosh, S.; et al. Utility of adiponectin as a biomarker predictive of glycemic efficacy is demonstrated by collaborative pooling of data from clinical trials conducted by multiple sponsors. Clinical Pharmacology and Therapeutics 2009, 86, 619–625. https://doi.org/10.1038/clpt.2009.88
- Lines 199-200. Provide a p-value.
Response: Data extracted from Table. P-values were not provided. No change.
At birth, preterm babies have low plasma adiponectin concentrations (n = 122; 19.9 ± 10.6 μg/mL) when compared to term babies (n = 75; 33.7 ± 13.6 μg/mL) [29].
- Lines 201,202,209,210. g/ml should be ug/ml. Please make sure all such values are correct throughout.
Response: Corrected.
Adiponectin is a robust biomarker of PPARγ activity. Existing literature on plasma adiponectin concentrations in preterm newborn babies suggest that PPARγ signalling may be underactive in preterm newborn babies. Kajantie et al. (2004) measured plasma adiponectin concentrations by ELISA in the cord vein of 197 infants [29]. Of them, 122 were born preterm (22 to 32 weeks’ gestation), and 75 at term (49 from a healthy and 26 from a diabetic pregnancy, with similar findings, and thus all data from term infants were pooled). At birth, preterm babies have low plasma adiponectin concentrations [n = 122; 3.7 (0.08 – 31.4) μg/mL] when compared to term babies [n = 75; 33.7 (5.5 – 54.8) μg/mL] [29]. Mean adiponectin concentrations increased from less than 1 μg/mL at 24 weeks’ gestation to approximately 20 μg/mL at term. Preterm females had 57% higher adiponectin concentrations (0 to 146%; P = 0.05) than preterm males. Adiponectin levels were lower in preterm infants with recent (<12 hours) exposure to maternal betamethasone but were unrelated to mode of delivery, preeclampsia, or impaired umbilical artery flow [29].
Hansen-Pupp et al. (2015) analysed adiponectin concentrations in cord blood at birth, and peripheral blood at 72 h, day 7, and then weekly until postmenstrual age of 40 weeks in 52 preterm babies born at 26 ± 1.9 weeks gestational age [30]. The mean adiponectin concentration increased from 6.8 ± 4.4 μg/mL at 72 h to 37.4 ± 22.2 μg/mL at three weeks. Mean adiponectin concentration during days 3 to 21 (21.4 ± 12 μg/mL) correlated with gestational age at birth (r = 0.46, P = 0.001), birth weight (r = 0.71, P = 0.001) and birth weight Standard Deviation Score (SDS) (r = 0.42, P = 0.003). Furthermore, mean adiponectin concentration during days 3 to 21 correlated with weight SDS, length SDS, and head circumference SDS at 35 weeks corrected gestational age (r = 0.62, 0.65, and 0.62, respectively; all P = 0.001). Peak concentrations at 3 weeks of age (n = 52; 37.4 ± 22.2 μg/mL) did not correlate with gestational age but positively correlated with catch-up growth (β = 0.021; CI:0.001– 0.041; P = 0.04) [30].
- Lines 206-217. Did adiponectin increase between what was measured in cord blood and at 72 h? Are there any data for postnatal changes in PPARγ among term and preterm infants?
Response:
Mean adiponectin concentration measured in cord blood was 3.7 [0.080–31.4] μg/mL (Kajantie et al)
Mean adiponectin concentration at 72 hours was 6.8 ± 4.4 μg/mL at 72 hours (Hansen-Pupp et al).
This suggests an increase from birth.
We have added Krishna et al. reference with data on preterm post mortem brain. To the best of our knowledge, no other data are available.
- Sections 4.2 and 4.3. Preterm infants are at risk of increased adiposity that is associated with increased risk of later obesity. Weight gain due to increased fat deposition will not be considered by many to be beneficial. Increasing fat-free mass would be better.
Response: We do not know the relative body composition in preterm babies that may or may not result with treatment. Further work is required.
- Section 4.5. Are there any data for the maternal-to-fetal transfer of pioglitazone? Such studies would need to use animal models with placental structures similar to those of humans. If PPARγ agonists are not transferred across the placenta to the fetus, the adverse response would be of maternal origin.
Response: Pioglitazone does cross the placenta. The following change was made:
Several animal studies have been conducted by Takeda Pharmaceuticals for the licensing of pioglitazone [38]. Pioglitazone is non-teratogenic in both rats and rabbits. High doses were associated with maternal toxicity (e.g., placental pathology, weight loss), implantation losses, reduced foetal size, live birth weight and survival ratios, and delayed pup development. It is suggested that the adverse effects on the offspring result from disturbances in maternal physiology rather than a direct toxic effect. Foetal growth restriction was attributable to the action of pioglitazone in diminishing the maternal hyperinsulinaemia and increased insulin resistance that occurs during pregnancy thereby reducing the availability of metabolic substrates for foetal growth rather than a direct toxic effect.
- Lines 339-340. This repeats lines 314-315.
Response: These lines have now been deleted.
A standard series of reproduction toxicity studies was conducted with pioglitazone in rats and rabbits. In fertility studies there was no effect on copulation, impregnation, or fertility index. Pioglitazone is non-teratogenic in both rats and rabbits. High doses were associated with maternal toxicity (e.g., placental pathology, weight loss), implantation losses, reduced foetal size, live birth weight and survival ratios, and delayed pup development. It is suggested that the adverse effects on the offspring result from disturbances in maternal physiology rather than a direct toxic effect. This is supported by differences in maternal and foetal NOAEL.
- Section 5. A key consideration for the preterm infant is organ immaturity that will influence metabolism, distribution, responses, and elimination of pioglitazone and other PPARγ agonists. Assessing those will require animal models that are more relevant to ELGAN that rodents.
Response: We agree that pharmacokinetics is an important aspect of future research. However, we think that this can be safely done in the preterm baby within the setting of an early phase dose escalation study.
- Section 6. The authors should consider mentioning the limitations of previous studies, such as the animal models, and suggest recommendations for what is needed to translate findings into clinical trials. This might also help to position them for such studies.
Response: Our aim is to progress to early phase dose escalation clinical trial with pharmacokinetic studies included. The manuscript already states the following:
We are currently developing formulation options for pioglitazone that will be suitable for administration to preterm babies. This will allow us to conduct early phase clinical trials to determine the safety and tolerance of pioglitazone in preterm newborn babies.
This has been amended to:
We are currently developing formulation options for pioglitazone that will be suitable for administration to preterm babies. This will allow us to conduct early phase clinical trials to determine the safety, and tolerance, and pharmacokinetics of pioglitazone in preterm newborn babies.